# Reputation Incongruence and the Preference of Stakeholder: Case of MBA Rankings

**DOI:** 10.3390/bs11010010

**Published:** 2021-01-19

**Authors:** Jin Suk Park, Mooweon Rhee

**Affiliations:** 1School of International Corporate Strategy, Hitotsubashi University, 2 Chome-1-2 Hitotsubashi, Chiyoda, Tokyo 101-8439, Japan; 2School of Business, Yonsei University, 50 Yonsei-ro, Seodaemun-gu, Seoul 120-749, Korea; mooweon@yonsei.ac.kr

**Keywords:** organizational reputation, preference theory, decision-making, organizational identity, MBA program

## Abstract

In this paper, we examine the effect of an organization’s multi-dimensional reputation on the external stakeholders’ preference for an organization in the notions of reputation incongruence. We propose that an organization’s incongruent reputation, or large variations among the reputations of each dimension, can be an unfavorable signal to its stakeholders based on theoretical ideas that claim reputation incongruence induces the ambiguity and risk of an organization perceived by stakeholders. We also investigate the moderating effect of reputation incongruence by positing that this incongruence may nullify the influences of reputation dimensions on the preferences of stakeholders. These propositions about reputation incongruence are empirically examined in the context of MBA programs of the global business schools which have three dimensions of reputation—career development, globalization, and research performance.

## 1. Introduction

Acquiring reputation as an intangible asset [1,2] is one of the most critical missions for organizations to increase their profits and mitigate their potential risks [3,4]. In the field of management and business, there have been consistent findings proving the benefits of organizational reputation that lead to an organization’s performance [5], which emphasizes the organizational reputation as a valuable, exceptional, and unique resource [6], and an important strategic asset [3]. Research on the multidimensionality of the organizational reputation deepens the understanding about how an organization can utilize its reputation as the important asset. An organization’s reputation involves a diverse collection of dimensions [7], so that the organization needs to concurrently consider and manage various reputation dimensions [8].

This study develops the ideas of the multidimensional reputation by turning focus to the portfolio or composition of reputation dimensions. We investigate the question of which patterns of a multidimensional reputation portfolio or composition can be perceived as favorable signals to key stakeholders, resulting in the desirable outcomes of organizations. Multi-attributes profiles such as organizational reputation can be described by two parameters, elevation, the averaged value calculated from each attribute, and *scatter*, the internal variability of the attribute values around the profile mean [9,10]. The elevation can incorporate the individual value of each reputation dimension, and the scatter can refer to variations among these individual values. While the positive effect of aggregated reputation has been well discussed [11], the variation among reputation dimensions is insufficiently recognized.

Thus, our first step of developing the framework for multidimensional reputation is to introduce the concept of reputation incongruence that denotes the incongruent or conflicting values between each reputation dimension. According to the definition of the scatter, an organization with good reputation in some dimensions and poor reputation in others can be described as an organization with incongruent reputation. Based on decision-making theories and reputation studies, we suggest that this reputation incongruence may function as a negative signal to key stakeholders because the signal is associated with ambiguity [12] and high risk [13] leading to the unfavorable reaction of stakeholders against the organization. We also argue that there is the moderating effect of reputation incongruence on the association between the elevation and the preference of stakeholders by positing that the averaged influences of reputation dimensions can be discounted by the larger variances among the dimensions. The reputation incongruence can decrease the reliability and importance of information about the elevation [14], so that stakeholders are likely to reduce the use of this unreliable information derived from the reputation. We expect the findings of this study can be applied to a broader range of empirical context, which is discussed in the conclusion.

## 2. Theory and Hypotheses

### 2.1. Multidimensionality of Reputation

A stream of research that is commonly labeled as *a composite view of a multidimensional construct* [15] focuses on the multidimensionality of organizational reputation based on a postulation that several reputation dimensions can compose an overall reputation of an organization. Weigelt and Camerer [2] propose that organizational reputation is a *set of economic and non-economic attributes* that belong to an organization. Fombrun et al. [7,16] also conceptualized corporate reputation as *a collective construct* consisted of various attributes or dimensions. Then, the researchers create the single index of overall reputation by averaging the individual values of reputation dimensions.

### 2.2. Effect of Elevation on the Preference of Stakeholders

Previous literature seems to comfortably assume that a favorable signal evokes the stakeholders’ favorable reaction because of the intuitive logical linkage between the favorable signal of overall high elevation and the increased preference of stakeholders [6,7,17]. The preference of stakeholders is a separate but closely follow-up step involving behavioral reactions after receiving a favorable signal (i.e., high elevation). For instance, the preference for the favorable reputation can be revealed by favorable behavioral reaction such as the willingness to pay more price [17], the actual selection of an option [18], the decision of social connection with an organization [19], and the actual amount of investment (e.g., time and money) to an option [20].

In the empirical context of this study, some media rankings for MBA programs also provide signal about whether a program is generally good or bad, and the stakeholders can react accordingly by aggregating the elevation of each reputation dimension or by just using the already-calculated overall rankings. The recent study of Igwe, Rahman, Ohalehi, Amaugo, and Anigbo [21] also demonstrated that students in business schools perceived different value dimensions such as career development and academic learning to develop overall attitude to the programs. The preference of prospective students, the reaction to the reputation of programs, can be represented by their application behaviors. For example, the prospective students can reveal their preference by applying to the reputable program. When it comes to another key stakeholder group, recruiters or employers, the preference of this group can be revealed in their recruiting behaviors such as the amount of offered salary. This stakeholder group is willing to offer a competitive salary to the graduates from the reputable schools to attract the graduates [22]. We, here, suggest a hypothesis about the elevation to extend the debate about the concept of scatter in the following section.

**Hypothesis** **1** **(H1).***The greater overall reputation an organization has, the more likely the stakeholders prefer the organization. (Elevation)*.

### 2.3. Effect of Scatter: Reputation Incongruence

The degree of scatter is an essential parameter providing a new arena for the debate about multidimensional reputation. To articulate the mechanism of this parameter’ effect on the preference of stakeholders in the context of reputation research, we introduce a new terminology, *reputation incongruence*, defined as a certain type of reputation signaling large variation in possible outcomes. In other words, the widely scattered value of reputation dimensions are a signal reflecting an organization has the mixture of pros and cons simultaneously, which can be specified by the terms of “higher variance in attribute values” [12].

We propose the reputation incongruence can increase *ambiguity* of the signal of an organization’s reputation based on the argument that stakeholders may have difficulty in estimating the overall quality of an organization from the ambiguous information in incongruent reputation. In other words, because reputation incongruence is a mixed signal including both positive and negative values, it is not clear for stakeholders whether this signal indicates a good or poor quality in general. This ambiguous signal seems to inform that there is a variation in actual qualities when the stakeholders select the organization with reputation incongruence. To examine the effect of this ambiguity on the preference of stakeholders, we elaborate the concept of an ambiguity in reputation incongruence and then the reaction of stakeholder to this ambiguity.

As Ellsberg [10] defines ambiguity as a status where decision makers do not know the exact likelihoods of outcomes from decision-making, a wider range of estimation can represent a situation with more ambiguity compared to a narrower range. The important mechanism of reputation incongruence, here, is to widen the range of estimation. Stakeholders are likely to use each value of dimension as the individual observation while estimating the overall quality of an organization, so that more variance among the value of dimensions can widen the prediction interval. For example, Jia, Luce, and Fischer [23] demonstrate that a within-option conflict (i.e., variance among attributes) serves as an important antecedent determining the width of a confidence interval that predicts the overall quality of a multi-attribute product.

The past literature on the topic of *ambiguity avoidance* implies that there is a substantial tendency of preferring an option with lower ambiguity to one with higher ambiguity [10]. This tendency can be explained by the high risk of the ambiguous signal because the inaccurate signal in reputation incongruence indicates that there is more possibility that the selection of an organization with the inaccurate signal can induce an unexpected *bad* result than the stakeholders predicted [24]. Situational contingents and individual differences, like the amount of rewards and risk-taking attitude, may have incidental effects on the attitudes and the behavioral patterns for the decision-making under the ambiguity; but in general, people are risk-aversive and strive to assure that the information they obtain is accurate enough to make a confident decision with the motivation of avoiding the potential adverse results or poor qualities [25].

Alongside the issue of an inaccurate assessment, the reputation incongruence also can act as a *conflicting* signaling from an organization that may produce an unpleasant dissonance that should be resolved by the stakeholders. Sensemaking theory supports the negative effect of reputation incongruence on the preference of stakeholders because the reputation incongruence can provoke stakeholders’ confusion about what the signal represents and cognitive dissonance about whether the organization should be chosen or not [26]. When the mechanisms of the conflicting signal and the inaccurate assessment are taken together, we can propose that the reputation incongruence of an organization entails an inaccurate assessment on the general performance of an organization [27] and the increased cognitive gaps induced by the sensebreaking [26]. Then, once stakeholders receive the signal, they view the organizations with incongruent dimensions less favorably than other organizations having a congruent reputation. Finally, by incorporating two parameters (i.e., elevation and scatter), we suggest the following hypotheses:

**Hypothesis** **2** **(H2).***The greater reputation incongruence an organization has, the less likely the stakeholders prefer the organization. (Scatter)*.

### 2.4. Moderating Effect of Reputation Incongruence

In addition to the main effect of reputation incongruence, we also suggest the moderating effect of reputation incongruence. Reputation incongruence is likely to diminish the main effect of elevation on the preference by inhibiting the stakeholders from using reputation as key information due to the *reduced reliability* of the signal information predicting the overall quality of an organization. As the ambiguous or unclear information derived from reputation incongruence seems to prompt the stakeholders’ suspicions about the trustworthiness of the signal [28,29], stakeholders may try to avoid using this incongruent information when they predict the quality of an organization. Thus, if the main effect of reputation incongruence is explained by the stakeholders’ suspicion about the overall quality of an organization, the moderating effect is accounted for by the suspicion about the quality of information. Moreover, we argue that the reputation incongruence can cause a randomness issue. If stakeholders recognize the widely distributed individual value of each dimension as a random observation, they are likely to interpret that the signal has a lower signal-to-noise ratio (SNR) that is less reliable information compared to a signal with higher SNR [29,30]. As Rhee, Kim, and Han [8] point out a *large variance problem* by stating “the amount of variation is positively correlated with the degree of randomness”, a fundamental assumption embedded in the debate about randomness in decision-making theory [8] supports that stakeholders could interpret the scattered observations as random incidents increasing the SNR of the signal.

The conventional idea about the reliability of measurements [31] is easily applicable to the arguments thus far. As more variances among items reduce the reliability of scale [9], reputation incongruence can reduce the reliability or the trustworthiness of its signal due to its incongruent values among each dimension. Eventually, the utility of the information from an incongruent reputation may be underestimated [32]. We propose the following hypothesis based on the rationales that reputation incongruence might reduce the overall reliability of information from organizational reputation:

**Hypothesis** **3** **(H3).***The interaction effect of reputation incongruence will serve to attenuate the main effect of elevation of reputation*.

## 3. Methods

### 3.1. Data and Samples

The empirical context of this study resides under Global MBA programs. As Rindova et al. [17] suggest that corporate recruiters, one of the most important stakeholders of MBA programs, heavily rely on the business schools’ reputations in order to reduce the uncertainty and asymmetry of information about the quality of business school graduates because the quality of MBA graduates is difficult to evaluate before hiring [30]. Thus, the context of our study is particularly appropriate for examining the effect of the distinct dimensions and the incongruent disposition of an organization’s reputation on the stakeholders’ reactions.

This study’s panel data set was gathered via two major sources: *Global MBA Rankings*, which is reported annually by the *Financial Times* [33], and *Best Business Schools*, which is also reported annually by the *Princeton Review*. The population in this study includes all MBA programs that were ranked in the top 100 lists of the *Financial Times’ Global MBA Rankings* [33] reports during the period from 2008 to 2016. 73 U.S. business schools and 72 non-U.S. schools were ranked at least once within these top 100 lists at least once during this period. *Best Business Schools* by the *Princeton Review* was also used in order to add more variables into this study. Due to missing information, we will use only overlapped observations between these two major data sources. After combining two data sources, 504 observations including 69 U.S. business schools and 25 foreign MBA programs from 2008 to 2016 were used in this study.

### 3.2. Dependent Variables

To examine our hypotheses, we employed two types of dependent variables measuring the preference of two different stakeholder groups. One measure represents the preference of recruiters (i.e., employers) revealed in recruiting behaviors and another measures refer to the preference of applicants (i.e., prospective students) for the MBA programs. First, the preference of employers as a key stakeholder group who recruits graduates from the MBA programs was measured by the first year’s salary they pay the graduates because the employers are willing to pay more for the graduates from reputable programs based on their preference about the programs [17]. In particular, the *Financial Times*’ MBA ranking [33] provides a weighted salary statistic that was calculated by “the average salary today with adjustment for variations between sectors” and the relative value of currencies between countries [34]. Due to the skewed distributions, we used the natural logarithm of this weighted salary. Secondly, to measure the preference of prospective students who consider enrolling in the MBA schools, we used the information of how many prospective students apply to an MBA program, which can be understood as how successfully the reputation can attract applicants. Because we also found the skewed distributions of this variable, we used the natural logarithm of the number of applicants.

### 3.3. Elevation of Reputation

Although there are several contending publications, we selected the *Financial Times* rankings [33] to measure the reputations of business schools because this rating system includes more various attributes and a larger number of global programs in comparison to other ranking systems. With this *Financial Times* public report [33], stakeholders can identify rankings in seven criteria: career progress, aims achieved, placement success, international mobility, international experience, doctoral program, and research productivity. In addition, the *Financial Times* ranking system [33] has a longer history, better name recognition, and arguably is more comprehensive than other contenders [35], providing appropriate measures for studying the multidimensionality of reputation.

To determine the major dimensions of the MBA programs’ reputation, we conducted confirmatory factor analysis with the various attributes reported in the *Financial Times* [33], based on the argument that the quality of business schools can be understood by the dimensions of research performance and effective career development [22] and the dimensions related to globalization or international diversity [36]. We categorized seven attributes used in the *Financial Times* ranking [33] into three major reputation dimensions of career development, globalization, and research performance. Three attributes—career progress, aims achieved, and placement success—were categorized into the career development dimension while the other two attributes—international mobility and international experience—were categorized into the globalization dimension. We also bounded the doctoral rank and the research rank as a factor in this confirmatory factor analysis. The result revealed that each attribute is significantly loaded on its bounded dimensions (RMSEA = 0.054, *p* < 0.001).

After the three dimensions of reputation had been verified, we averaged the rankings of the attributes for each dimension to calculate each dimensions’ ranking score. Next, we calculated reversed indices from this score of each dimension’s ranking because the lower score of ranking refers to the better reputation (i.e., being ranked in 1 means the best reputation compared to being ranked in 100), which can cause confusion during the interpretation of results. To do this, we subtracted the ranking scores from 101 since that the lowest possible ranking in our dataset would be 101. Next, we calculated the logarithm of this reversed variable likewise other variables. Thus, a dimension’s reputation, D*_j_*, of an MBA program, *i*, in the selected current year, *t*, is formally defined with the average ranking of each attribute, A*_ijk_*, as following:D*_ijt_* = log(101 − ∑A*_ijkt_*/N)(1)
where:D*_ijt_*: is *j*th dimension’s reputation of *i-*th MBA program in the selected current year, *t.*A*_ijk_*: is a *k*th subcategory’s ranking of aggregated measure for *j*th dimension’s ranking.

Then, we calculated a composite measure by averaging the weighted elevation of each dimension, according to the regression procedure of weighting multi-attribute options [24]. Based on the results from pretests using GEE regression models, we identified weights for each dimension: For the preference of job applicants, the career development was weighted by the coefficient of 0.10 (*p* < 0.01), globalization by 0.05 (*p* < 0.01), and research performance by 0.18 (*p* < 0.01) while for the preference of recruiters, career development by 0.03 (*p* < 0.01), globalization by 0.04 (*p* < 0.01), research performance by 0.04 (*p* < 0.01). Thus the overall elevation of reputation, OE*_it_*, could be calculated by $\sum_{j=1}^{3}$
*w_itj_**D*_ijt_*, where *w_j_* denotes the weight. Finally, we centered OE*_it_* by mean in order to avoid multicollinerity with reputation incongruence and interaction term with it [37].

### 3.4. Reputation Incongruence

In this paper, the reputation incongruence is defined as variations among the perceived reputation of each dimension. To measure this variable, we used the coefficient of variation (V), which is “the standard deviation divided by the mean” [38], calculated from the weighted elevations of three reputation dimensions. Many organizational studies have adopted this V as a statistical measure indicating the internal variability in social groups such as top management teams, task groups, or organizations on numerous dimensions [39]. Thus, in this study, the higher value of V denotes that eventually there will be a more scattered reputation and more incongruence in reputation. This variable was also used in natural logarithmic form, and then centered to reduce the multicollinearity with the individual values of reputation dimensions during the interaction analysis.

### 3.5. Control Variables

***U.S./non-U.S.*** It has been discussed that U.S. higher educational programs including MBA programs are more appealing to non-U.S. employers and students as well as stakeholders probably in the U.S., compared to foreign MBA programs [36]. we created a dummy variable indicating whether a school is located in the U.S. (1) or not (0) to identify this difference in nationality by using information in the *Princeton Review*.

***Public/private.*** Resource dependence theory suggests that private schools, which are more dependent on students and businesses for resources, should devote more time and effort into MBA programs to immediately meet these stakeholders’ needs [40]. On the other hand, because public institutions have some minimum level of guaranteed state funding, they can distribute their attention to other less profitable activities than MBA programs [40]. Thus, we included dummy variable that denotes 0 for public school and 1 for private based on information from the *Princeton Review*.

***Annual tuition*.** The prospective students strive to maximize the difference between what they paid for their MBA programs and what they receive after graduation [34]. This variable was collected from the *Princeton Review* and used in natural logarithm to avoid skewness.

***Size of MBA program*.** Organizational size is an important factor that has been found to affect the stakeholders’ perceptions and preferences of an organization by linking the size with the organizations’ higher visibility, more power, and legitimacy [41]. We operationally defined the size of each MBA program as the number of annual enrollments to the program by using the data of the *Princeton Review* and other supplementary sources such as program-specific brochures and web-pages. Because of the skewness, this variable was also used in natural logarithm form.

***Geographic location of business schools*.** This factor influences the students’ preferences because of the living costs [34] and the recruiters’ preferences to the business schools due to the accessibility [42]. We created three dummy variables to control for the effect of geographical locations based on the information of four categories about the business schools’ location provided by the *Princeton Review*: village, town, city, and metro.

***Age of MBA program.*** An organization with a long history usually has a competitive advantage over other organizations with a shorter history because of the stable membership that develops favorable relationships with other stakeholders and deepens the understanding about the complex social system [43]. This advantage of long history or age can be directly connected to the perception of the general audiences including the prospective students and the recruiters [44]. This age variable was controlled in natural log form due to its skewness, which was shown in the *Princeton Review*.

***Weighted salary.*** One of the prospective students’ main concerns is how they can increase their salary after graduation [34,42], we included this variable in the models using the preference of the students as the dependent variable. Because this variable was already used as the dependent variable in the models for the preference of recruiters and the correlation between current year’s weighted salary and the lagged year’s one was extremely high (*r* = 0.95), we excluded this in the latter regression models.

***Characteristics of student body.*** Previous work experience before enrollment to the MBA programs are expected to positively associate with a higher post-graduation salary because recruiters assume that longer work experience implies more maturity and wisdom of graduate students from the MBA programs [45]. Likewise, the GMAT score is an important criterion where recruiters evaluate the quality of MBA students [17,45]. Lastly, we controlled the ratio of female students in MBA programs due to the gender differences in salary after graduation [46] and the damaged image in the parity of the programs [45]. Each observation’s work experiences and GMAT scores were collected from the *Princeton Review* while the ratio of female students was from the *Financial Times* [33]. All of these variables were used in natural logarithm.

### 3.6. Models

If P*_it_* denotes the stakeholders’ preference about an MBA program, *i*, in a focal year, *t*, as a lagged response to the reputation of the programs in a previous year (*t*−1), we model the following equation to provide models to test our hypotheses in this study:P*_it_* = αOE*_i_*_(*t*−1)_ + βRI*_i_*_(*t*−1)_ + γOE*_i_*_(*t*−1)_RI*_i_*_(*t*−1)_ + δX*_i_*_(*t*−1)_ + μ*_it_*
where:OE*_i_*_(*t*−1)_: is (*t*−1) year’s overall reputation of *i*-th MBA program.RI*_i_*_(*t*−1)_: is (*t*−1) year’s reputation incongruence of *i*-th MBA program.X*_i_*_(*t*−1)_: is (*t*−1) year’s control variables of *i*-th MBA program.μ*_it_*: is (*t*−1) year’s intercept of *i*-th MBA program.

Here, OE*_i_*_(*t*−1)_ and RI*_i_*_(*t*−1)_ refer to each independent variable where *t*, the year of observation, ranging from 2008 to 2016. Specifically, OE denotes the overall reputation or elevation and RI the reputation incongruence among these three dimensions (see the measure section) while X*_i_* denotes control variables. To estimate the parameters, we separately used recruiters’ preference (i.e., log[weighted salary(*t*)]) and prospective students’ preferences (i.e., log[number of application(t)]) as two different dependent variables. In other words, there were two different analyses using each dependent variable. The equations also included the interaction terms to test our hypothesis 3. The negative effect of these interaction terms would imply reduced main effects of each dimension by assuming that each reputation dimension has a positive association with the preference of employers.

The parameters of each equation will be estimated using an unbalanced panel data set with yearly time periods. We use the generalized estimating equations (GEE) model to analyze population average, which is less likely to be misled by the assumption on distribution [47,48]. To account for autocorrelation among an MBA programs’ observations, we adopt unstructured correlation matrices due to the difficulties in assuming the correlations. The summary of variables is shown in Table 1.

## 4. Results

Table 2 and Table 3 separately report descriptive statistics and correlations for variables used in the models for the preference of the prospective students and recruiters. Because of missing information for the number of applicants, the models for the prospective students group had the slightly smaller numbers of observation (N = 465) than the models for the recruiters (N = 504). Of particular interest is the negative correlation between the preference of these stakeholders and reputation incongruence, corresponding to a major hypothesis of this study. The preference of prospective students (i.e., the number of applicants) was significantly correlated with the reputation incongruence and the preference of recruiters (i.e., weighted salary) was also with the reputation incongruence. Among several independent and control variables, there were significant correlations, too. Especially, the reputation dimensions were correlated with many other variables. Variance-inflation factors in every regress model, however, were less than 2.87 in the models used the prospective students’ preference as the dependent variable, and less than 1.72 in the models used the recruiters’ preference, respectively, and any correlation coefficient did not exceed 0.80 among the variables, so that there was very little probability of a multicollinearity problem [49].

Table 4 shows the results from the GEE estimates of the Equation (1) using the number of applicants as the dependent variable. In Model 1, the base model with containing only the control variables reports that locating in U.S., future salary, large size, and high GMAT score were positively associated with the 1-year lagged number of applicants while the expensive tuition and the location in town were negatively associated with the dependent variable.

In Model 2-1, we added the effects of overall reputation and reputation incongruence to test Hypothesis 1 and 2. The positive coefficient for the overall reputation was significant (β = 0.9083, *p* < 0.01) and the negative one for the reputation incongruence also significant (β = −0.4706, *p* < 0.01), so both of the hypotheses were supported in this analysis. In Model 2-2, we included the interaction term between the two dependent variables to test Hypothesis 3. Model 2-2 displays the significant coefficient for this interaction term with a negative direction (β = −4.0988, *p* < 0.01), which supported the prediction of Hypothesis 3. Statistically significant improvements in Wald chi-square statistics from Model 1 to Model 2-1, and from Model 2-1 to Model 2-2 also showed that adding the main and moderating effects of reputation incongruence considerably improves the fit of the models. Figure 1 illustrates the moderating effect found in Model 2-2, and the patterns of other models for prospective students were similar with this.

When it comes to another stakeholder group (i.e., recruiters), the structure of modeling is the same as the prospective students group (see Table 5). Model 3 was conducted as the base model to examine the effect of control variables on the 1-year lagged weighted salary, and Model 4-1 was used to estimate the coefficients for overall elevation and reputation incongruence. Model 4-2 was for the interaction between these two independent variables. With respect to the controls, Model 3 shows that locating in the U.S., operating as a private school, having a more expensive tuition, and possessing a longer history were significantly associated with a higher salary while schools’ locations in villages and towns were in the lower salary (see Table 5). As predicted, the results in Model 4-1 and 4-2 supported Hypothesis 1 and 2 by displaying the significant coefficient for overall elevation (β = 0.0014, *p* < 0.01), reputation incongruence (β = −0.0127, *p* < 0.01), and the interaction term (β = −0.0015, *p* < 0.01). There were also the significantly improved Wald chi-square statistics between these models. Thus, the Hypothesis 1, 2, and 3 were supported in this analysis.

Figure 2 depicts these patterns of moderating effect in the models using the weighted salary as a dependent variable.

## 5. Conclusions

This study examined the influence of elevation and scatter in multi-dimensional reputation that can establish the pattern of a reputation portfolio. We found the positive effects of the overall and individual elevations in each dimension on the preference of the stakeholders as visualized in Figure 3. Before concluding that the high elevations (or good reputations) are good, this study revealed another important determinant of favorable reputation, reputation incongruence. Even though the good reputation seems to attract key stakeholders, more variance among the elevations of dimensions can lessen the favorability of the signal. The reputation incongruence by itself not only seems to deliver the unfavorable messages from an organization to its stakeholders, but also to dilute the positive effect of the high elevations.

This study’s major contribution to reputation theory is to introduce and include this concept of reputation incongruence that has been out of the picture as a key issue for academic research. Although, there have been a few publications using similar terminologies with *reputation incongruence*, most of them only examined time-based inconsistency rather than the variance among reputation dimensions. For example, Hannan and Freeman [50] refer to the unreliability and unaccountability of an organization when this organization shows inconsistent performance or suddenly produces exceptionally poor quality products. Parker and Krause [51] also used the term, reputation incongruence, in this context in order to describe an unusual performance of an organization compared to its past-accumulated performance. This kind of over time incongruent performance that affects the reputation of an organization has been found by the well-developed research stream of aspiration level and performance feedback [52,53].

Meanwhile, the major findings of this study can be addressed in the respects of two seemingly competing theories: portfolio investment theory and diversification theory. Alongside the ambiguity issue, stakeholders may also avoid an incongruent multi-attributes option because the investment on the option seems a higher risk than a congruent one. Reputation incongruence can be interpreted as a concentrated investment portfolio that is a riskier choice than a diversified investment portfolio [54,55]. Polkovnichenko [56] demonstrates that households that own the concentrated portfolios of individual stocks are well aware of the higher risk associated with such investments compared to alternative investment products that have diversified portfolios. We reason that the high risk of a concentrated investment portfolio is linked to reputation incongruence by assuming that the stakeholders’ choice of an organization with reputation incongruence is a concentrated investment on the limited numbers of reputable dimensions. Or selecting an organization with reputation incongruence can be analogous to purchasing an investment product with a concentrated portfolio. If the concentrically invested dimension turns out to be a false signal, the investors (i.e., stakeholders) are likely to lose a large portion of their return on investments concentrated on the reputable dimensions. On the other hand, when the value of reputation is evenly distributed over several reputation dimensions, stakeholders can have backup plans even if a few dimensions fail or fall short of their expectations. Stakeholders may also easily have access to information about this reputation portfolio of an organization because many media ranking systems such as Fortune 500′s World Most Admired Companies [57], publicly report the separate rankings of each reputation dimension. From this signal information, stakeholders may be able to capture the underlying risk of the reputation portfolios presenting the reputation incongruence.

This prediction from portfolio theory seems to presumably contradict the suggestion of diversification theory [58] arguing a firm’s risk will be reduced when it practices a constrained diversification strategy that concentrates its resources on a single business sector among related businesses. This perspective, however, is more appropriate to the situation of when an organization expands to a new business rather than when stakeholders evaluate an existing reputation portfolio of the organization. The diversification theory suggests that managers may want to exploit their organizations’ internal strengths by expanding business to related and similar sectors with the expectation of synergy between well-performing current businesses and new ones [59]. In this case, it may be a risk-aversive choice to distribute a relatively small amount of resources to a new business to avoid the enormous damage from the possible failure of the new business. However, the reputation dimensions of an organization are not business sectors that can be strategically selected or deselected by the organization. Instead, the concept of a reputation dimension is closer to a given criterion [1] where all actors in the same industry should be evaluated. For example, the reputation of Apple, a computer manufacturing company, can be evaluated by the elevation and scatter of its reputation dimensions such as innovativeness, financial performance, or social responsibility while other competitors in the same industry are evaluated under the same dimensions [57]. Then, stakeholders can identify whether Apple has a diversified reputation portfolio or a concentrated one by considering the risk of reputation incongruence. However, this computer manufacturing company’s radical decision of crossing the border to the unrelated mobile device industry [60] does not necessarily indicate that this company has the risk of the reputation incongruence.

Although few studies clearly elucidate the definition of the reputation dimension, previous literature investigating the multidimensionality of reputation heavily relies on the theories developed in the topic of multi-attributes rather than diversification theory concerning the multi-sectors. It is easy to witness that the literature of multidimensionality consistently uses the term of “attribute” to refer to the reputation dimension [7,11], which suggests that there is a significant overlap between these two concepts. Rosen [61] (p.6) describes workers in labor market as a multi-attribute option “prepackaged with various combinations of skills and traits [attributes], some productive and others counterproductive” and states “employers cannot detach the less desirable ones [attributes] from any single worker.” Thus, reputation dimension seems to serve as an inherent attribute that can be combined and evaluated rather than an external business sector that can be selectively chosen.

Furthermore, the need for a clear comparison between the qualities of organizations seem to induce universally applicable reputation dimensions in the same industry, which means that any organization is barely free from the given evaluation criteria determined by the dimensions. As mentioned earlier, one of the key benefits of reputation is to provide transparent information to help external stakeholders overcome information asymmetry between the stakeholders and an organization [62]; and the stakeholders can obtain more precise information from reputation by comparing a target organization’ reputation and other contenders’. For example, corporate social responsibility became an essential evaluation criterion to measure any for-profit firm’s performance in many countries [63]. Therefore, our study has more potential to be extended in the perspective of portfolio theory rather than a strategic diversification while not limiting its contribution just on MBA programs and educational sectors.

For the managerial suggestions, because organizational reputation is an important competitive advantage of an organization [64] and communication channel to key stakeholders [65], it needs to understand and manage the organizational reputation. It seems straightforward for managers that possessing a good reputation in one dimension or in general can improve the key stakeholders’ evaluation on their organizations. However, if managers need to concurrently handle the several dimensions of reputation and to allocate limited resources across these dimensions, they need more clues than just “Good reputation is good”. The implication of this study that emphasizes less variation among each reputation dimensions can provide a better reputation portfolio in general can be applicable to every activity of managing organizational reputation such as building, maintaining, and repairing the reputation [66].

Regardless of such contributions, this study has its own limitations, too. First, the empirical tests and the results were utterly from the specific context of global MBA programs. The findings in this empirical context need to be replicated and expanded in other settings, although the environment of this industry now resembles the ones of other for-profit industries [67] and the MBA programs, like companies in the business world, become more strategic in competition with other programs [68], Some industries such as the automobile and pharmaceutical industries have reliable regulatory agencies serving as alternative information sources for stakeholders other than reputation. The most intriguing venue of future study inspired by the recent phenomenon may be an impact of COVID-19 on behavioral patterns dealing with reputation incongruence. This will also add more practical values to the current findings. Yue, Gizem Korkmaz, & Zhou [55] found that there was a risk-aversive tendency by people who experienced the virus outbreak more personally. More sensitive reaction of stakeholders to an incongruence option may be observed depending on environmental threats and/or personalized trauma. Organizational work environment can be also considered as an empirical setting that benefits from this study’s finding. Imposing conflicting job expectations would induce employees’ job satisfaction by increasing role ambiguity [69]. Thus, it is worthwhile to investigate an incongruence signal for different tasks and its impact on employees’ behaviors and attitudes. Second, we didn’t give an attention to potential difference between two stakeholder groups (i.e., recruiters versus prospective students). Future research may be able to include the possible influence of the stakeholder groups’ various characteristics in order to investigate the effect of reputation incongruence given that the multidimensionality is usually discussed in the perspective of multiple stakeholders such as job seekers, investors, or financial analysts [7,16]. Third, the antecedent of reputation incongruence can be an important topic for future research while our study’s focus was merely on the outcomes of incongruent reputation. The amount of resources retained by an organization is promising the candidate for the antecedent. As an organization consumes and allocates its available resources to manage its reputation [7,70], resource allocation studies can be a theoretical backdrop suggesting the size or available resource influences the pattern of resource distribution within an organization [71].

## Figures and Tables

**Figure 1 behavsci-11-00010-f001:**
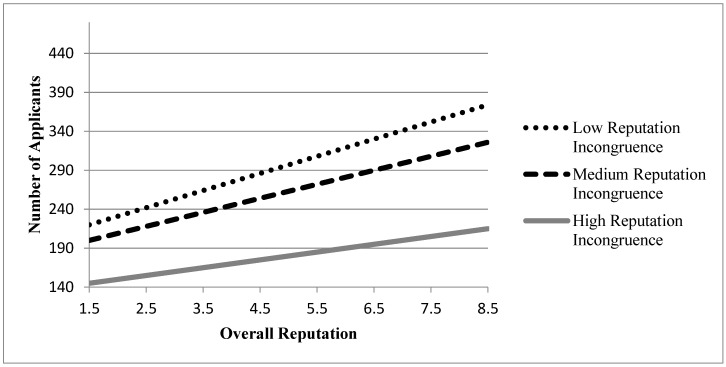
Moderating effect of reputation incongruence in Model 2-2. Note: High/Low groups were determined by cases outside 1 S.D.; Medium group was determined by cases within 1 S.D.

**Figure 2 behavsci-11-00010-f002:**
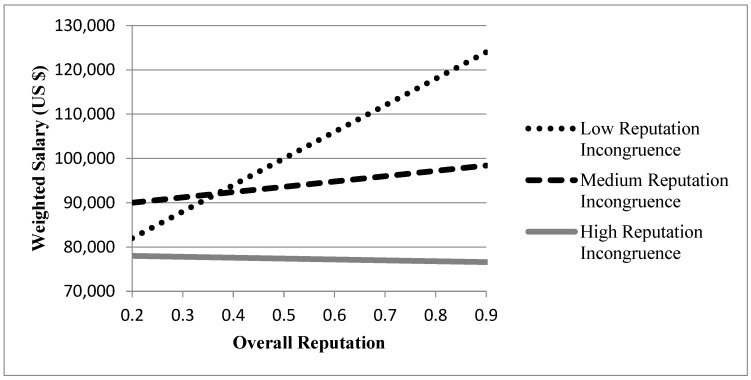
Moderating effect of reputation incongruence in Model 4-2. Note: High/Low groups were determined by cases outside 1 S.D.; Medium group was determined by cases within 1 S.D.

**Figure 3 behavsci-11-00010-f003:**
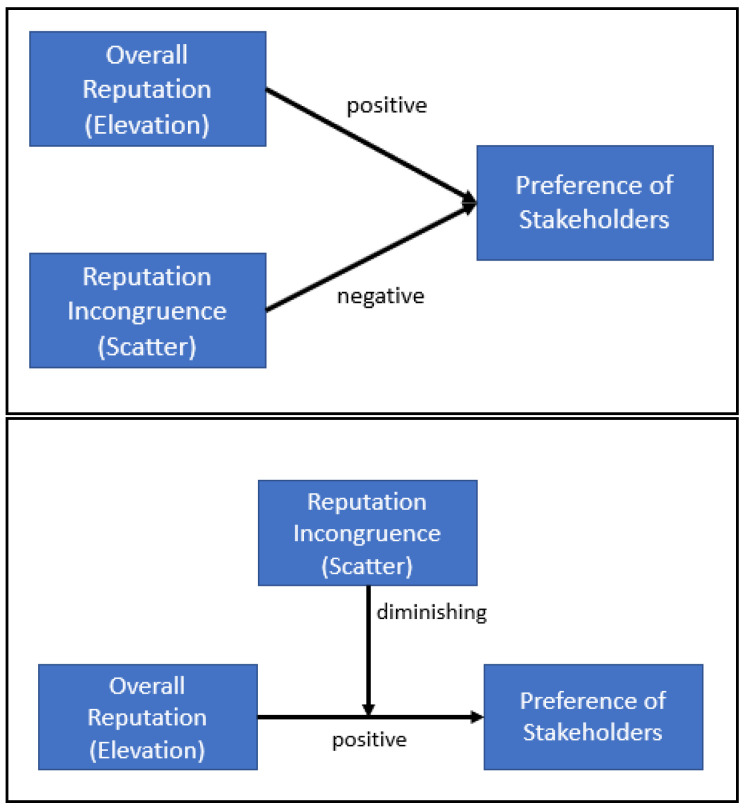
Nomological network of the study’s findings.

**Table 1 behavsci-11-00010-t001:** Summary of variables included in the GEE models.

Variable		Measure	Source	Model
Dependent variable	Preference of recruiters	First year salary of graduates	Financial Times	For recruiters
	Preference of prospective students	Number of annual applications	Princeton Review	For students
Independent variable	Overall reputation	Aggregated reputation from rankings	Financial Times	For both
	Reputation incongruence	Variation of reputation among sub-rankings	Financial Times	For both
Control variables	Nationality of a program	U.S. or non-U.S. based	Princeton Review	For both
	Public entity	Public or private program	Princeton Review	For both
	Annual tuition	Tuition in U.S. dollar	Princeton Review	For both
	Size of a program	Number of annual enrollments	Princeton Review	For both
	Location of a program	Village, town, city, or metro area	Princeton Review	For both
	History of a program	Age of a program	Princeton Review	For both
	Weighted salary	Salary of graduates (excluded in analysis)	Financial Times	For recruiters
	Current students’ work experience	Average working years before a program	Princeton Review	For both
	Current students’ female ratio	Reported ratio of female to male students	Financial Times	For both
	Current students’ GMAT score	Average GMAT score	Princeton Review	For both

**Table 2 behavsci-11-00010-t002:** Descriptive statistics and correlation coefficient in the models for prospective students (N = 495).

		Mean	S.D.	Min	Max	1	2	3	4	5	6	7	8	9	10	11	12	13	14	15
1.	Log(number of applicants)	2.98	0.42	1.63	3.96	1														
2.	U.S. school	0.56	0.50	0	1	0.33 **	1													
3.	Private	0.45	0.50	0	1	0.32 **	0.21 **	1												
4.	Village	0.06	0.23	0	1	−0.05	0.13 **	0.07 *	1											
5.	Town	0.11	0.32	0	1	−0.13 **	0.11 **	−0.14 **	−0.09 *	1										
6.	City	0.32	0.47	0	1	0.03	0.08 *	−0.08 *	−0.16 **	−0.24 **	1									
7.	Log(annual tuition)	4.47	0.17	3.79	4.86	0.29 **	−0.11 **	0.31 **	0.09 *	−0.13 **	−0.04	1								
8.	Log(weighted salary)	5.03	0.09	4.49	5.26	0.74 **	0.21 **	0.35 **	0.08 *	−0.01	0.04	0.54 **	1							
9.	Log(program size)	2.21	0.32	1.48	3.00	0.67 **	0.31 **	0.39 **	−0.05	−0.14 **	−0.01	0.30 **	0.60 **	1						
10.	Log(history)	0.00	0.29	−1.72	0.55	0.14 **	0.48 **	0.09 *	0.06	−0.06	0.04	−0.02	0.05	0.20 **	1					
11.	Log(work experience)	0.70	0.10	0.30	1.00	−0.03	−0.61 **	−0.17 **	0.00	−0.06	−0.14 **	0.18 **	0.00	−0.16 **	−0.34 **	1				
12.	Log(Female ratio)	1.46	0.11	0.85	1.75	0.19 **	0.07 *	−0.04	−0.07 *	−0.10 **	−0.03	0.20 **	−0.01	0.18 **	0.03	0.03	1			
13.	Log(GMAT score)	2.82	0.02	2.72	2.86	0.68 **	0.21 **	0.20 **	−0.03	−0.09 *	0.15 **	0.32 **	0.67 **	0.52 **	0.06	−0.09 *	0.06	1		
14.	Overall reputation	0.00	0.46	−1.54	0.67	0.58 **	0.46 **	0.03	−0.04	0.01	0.12 **	0.13 **	0.42 **	0.56 **	0.20 **	−0.20 **	0.12 **	0.45 **	1	
15.	Reputation incongruence	0.00	0.06	−0.03	0.77	−0.15 **	0.33 **	0.03	−0.01	0.04	0.14 **	−0.25 **	−0.05	−0.08 *	0.16 **	−0.27 **	−0.10 **	−0.01	0.11 **	1

note: * *p* < 0.05, ** *p* < 0.01.

**Table 3 behavsci-11-00010-t003:** Descriptive statistics and correlation coefficient in the models for recruiters (N = 504).

		Mean	S.D	Min.	Max.	1	2	3	4	5	6	7	8	9	10	11	12	13	14
1.	Log(weighted salary)	5.02	0.09	4.49	5.26	1													
2.	U.S. school	0.56	0.50	0	1	0.21 **	1												
3.	Private	0.45	0.50	0	1	0.35 **	0.21 **	1											
4.	Village	0.06	0.23	0	1	−0.08 *	0.13 **	0.07 *	1										
5.	Town	0.11	0.32	0	1	−0.01	0.11 **	−0.14 **	−0.14 **	1									
6.	City	0.32	0.46	0	1	0.04	0.08 *	−0.08 *	−0.08 *	−0.16 **	1								
7.	Log(annual tuition)	4.47	0.17	3.79	4.86	0.54 **	−0.11 **	0.31 **	0.31 **	0.09 *	−0.13 **	1							
8.	Log(program size)	2.21	0.32	1.48	3.00	0.60 **	0.31 **	0.39 **	0.39 **	−0.05	−0.14 **	−0.01	1						
9.	Log(History)	0.00	0.29	−1.72	0.55	0.05	0.48 **	0.09 *	0.09 *	0.06	−0.06	0.04	−0.02	1					
10.	Log(work experience)	0.70	0.10	0.30	1.00	0.00	−0.61 **	−0.17 **	−0.17 **	0.00	−0.06	−0.14 **	0.18 **	−0.16 **	1				
11.	Log(Femaleratio)	1.46	0.11	0.85	1.75	−0.01	0.07 *	−0.04	−0.04 *	−0.07 *	−0.10 **	−0.03	0.20 **	0.18 **	0.03	1			
12.	Log(GMAT score)	2.82	0.02	2.72	2.86	0.67 **	0.21 **	0.2 **	0.20 **	−0.03	−0.09 *	0.15 **	0.32 **	0.52 **	0.06	−0.09 *	1		
13.	Overall reptuation	0.00	13.5	−36.9	35.12	0.58 **	−0.07	0.25 **	−0.05	−0.09 *	0.00	0.4 **	0.59 **	0.02	0.11 **	0.05	0.57 **	1	
14.	Reputation incontruence	0.00	0.57	−3.57	1.23	−0.26 **	0.09 *	−0.08 *	−0.08 *	−0.12 *	−0.01	−0.07 *	−0.21 **	−0.21 **	0.18 **	−0.11 **	−0.08 *	−0.30 **	1

note: * *p* < 0.05, ** *p* < 0.01.

**Table 4 behavsci-11-00010-t004:** GEE estimates of the number of applicants, 2008–2016.

Variable	Model 1	Model 2-1	Model 2-2
U.S. school	0.2622 **	0.2253 **	0.1927 **
	(0.0641)	(0.0504)	(0.0522)
Private school	0.0294	0.0266	0.0730
	(0.0485)	(0.0307)	(0.0295)
Village	−0.0773	−0.0499	−0.0517
	(0.0600)	(0.0472)	(0.0490)
Town	−0.1120 *	−0.0624	−0.0280
	(0.0573)	(0.0427)	(0.0440)
City	−0.0476	0.0535	0.1268 **
	(0.0450)	(0.0316)	(0.0319)
weighted salary	1.2450 **	1.3621 **	0.3510 *
	(0.2089)	(0.1927)	(0.1782)
Annual tuition	−0.1840 *	−0.1492 *	0.0454
	(0.0842)	(0.0631)	(0.0551)
Program size	0.1319 **	0.3068 **	0.4916 **
	(0.0337)	(0.0408)	(0.0398)
Work experience	0.1272	0.0020	−0.1253
	(0.0865)	(0.1062)	(0.1004)
Female student	−0.0748	−0.0315	−0.0882
	(0.0752)	(0.0698)	(0.0571)
GMAT	4.6125 **	4.1258 **	3.0295 **
	(0.5222)	(0.5110)	(0.4491)
History	0.0217	0.1242	0.1261
	(0.1058)	(0.0682)	(0.0669)
Overall elevation (H1)		0.9083 **	1.2400 **
		(0.2933)	(0.2530)
Reputation Incongruence (H2)		−0.4706 **	−0.6422 **
		(0.1422)	(0.2431)
OE X RI (H3)			−4.0899 **
			(1.6115)
Constant	−1.6030 **	−1.5669 **	−0.8642 **
	(0.1608)	(0.1380)	(0.1291)
Wald Chi-square	289.17 **	685.11 **	711.97 **
D.f.	11	13	14
N	465	465	465
VIF	1.92	2.70	2.87

Note: * *p* < 0.05, ** *p* < 0.01; Standard errors are in parentheses; OE denotes overall elevation; RI denotes reputation incongruences.

**Table 5 behavsci-11-00010-t005:** GEE estimates of weighted salary, 2008–2016.

Variable	Model 6	Model 7-1	Model 7-2
U.S. school	0.0423 *	0.0012	0.0454 **
	(0.0208)	(0.0221)	(0.0204)
Private school	−0.0581 **	−0.0288 *	−0.0893 **
	(0.0140)	(0.0155)	(0.0152)
Village	−0.0280 *	0.0010	0.0236
	(0.0148)	(0.0189)	(0.0193)
Town	−0.0265 *	0.0072	0.0178
	(0.0124)	(0.0142)	(0.0173)
City	0.0034	0.0155	0.0430 **
	(0.0079)	(0.0106)	(0.0105)
Annual tuition	0.0336 **	0.0173	0.0518 **
	(0.0119)	(0.0122)	(0.0149)
Program size	0.0218	0.0604 **	0.0482 **
	(0.0121)	(0.0127)	(0.0120)
History	0.0871 **	−0.0013	0.1025 **
	(0.0328)	(0.0376)	(0.0355)
Work experience	−0.0201	0.0016	0.1012 **
	(0.0273)	(0.0242)	(0.0283)
Female student	−0.0160	−0.0081	−0.0207
	(0.0217)	(0.0166)	(0.0214)
GMAT	0.0525	0.2980 **	0.0469
	(0.1281)	(0.0961)	(0.1411)
Overall elevation (H1)		0.0014 **	0.0007 **
		(0.0002)	(0.0003)
Reputation Incongruence (H2)		−0.0127 **	−0.0145 **
		(0.0041)	(0.0038)
OE X RI (H3)			−0.0015 **
			(0.0003)
Constant	0.4814 **	0.5886 **	0.4774 **
	(0.0345)	(0.0274)	(0.0386)
Wald Chi-square	57.79 **	107.48 **	165.85 **
D.f.	11	13	14
N	504	429	429
VIF	1.44	1.72	1.70

Note: * *p* < 0.05, ** *p* < 0.01; Standard errors are in parentheses; OE denotes overall elevation; RI denotes reputation incongruences.

## Data Availability

The original data source entities do not allow the reproduction and representation of the raw data in other distribution channels and/or platforms. A direct contact to the information publishers is recommended.

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
