# Peer review of "Reputation Incongruence and the Preference of Stakeholder: Case of MBA Rankings"

_behavsci, 2021, doi:10.3390/bs11010010_

Round 1

Reviewer 1 Report

The article describes the issue that has not been widely researched so far. Although the importance of overall organization’s reputation for stakeholders is rather obvious (Hypothesis 1), examining the impact of its incongruence on the preferences of stakeholders is more complex. The obtained results confirm of what intuition suggests (confirmation of Hypothesis 2). The presented analysis of reports on MBA programs expands the way of looking at the issue of reputation and, as the authors noted, may constitute the basis for further research, e.g. in other industries, taking into account various groups of stakeholders, in combination with other factors influencing the preferences of stakeholders, etc.

I have no substantive comments to the article, because the authors properly present and analyse the studied issue. The research methodology is also understandable presented, although I would suggest to list the variables used in the formulas under the formulas, as usually authors do.

The second suggestion is to include a table containing all the variables, their descriptions, and attributes. Such table would make understanding of methodology and results easier.

The research hypotheses were clearly confirmed.

Graphical presentation of the results of the analysis is well chosen and increases the value of the article. There is only missing a clear explanation of criteria used for classifying the reputation incongruence as low, medium and high  (I’m referring to the moderating effect of reputation incongruence presented in figures 1 and 2)

Conclusions are a good summary of the results obtained and indicate their usefulness in managing the organization's reputation, as well as for future research in this area.

As for the references, most of the publications are from the early 2000s, but there are also some much older. Their selection can be partly explained by the fact that the concept of organizational reputation isn’t new, but references to publications over the past few years would be good for overall evaluation of the paper.

Summary of suggestions and comments:

  1. The authors should consider listing the variables used in the formulas directly under the formulas.
  2. The authors should consider including a table with all variables.
  3. The authors should present criteria of determining the low, medium and high reputation incongruence.
  4. References to more current publications would be welcomed.
  5. In line 242 the formula is not displayed correctly - the sum symbol is missing (this may be the result of converting the document to PDF)
  6. In line 258 the order of words is wrong - "employers and as students well as stakeholders", should be "employers and students as well as stakeholders" probably.
  7. After table 2, the line numbering is missing until the end of the text

Author Response

Dear Reviewer 1,

We would like to thank you for the thorough and constructive review for the paper. It was also greatly appreciated to have a chance to improve the initial draft and resubmit the current version of the manuscript. Without your positive feedback and helpful comments, this would not be possible at all. The attached file is the revised manuscript in a Word format.

Let us address all the issues you pointed out with the resubmitted manuscript as follow:

  1. The authors should consider listing the variables used in the formulas directly under the formulas.
  • The lists were added right below each equation. (Line 238 to 240; Line 320 to 324)
  1. The authors should consider including a table with all variables.
  • The sentence, “The summary of variables is shown in Table 1” (Line 339 and 340) and the table (Line 357 to 359) were added.
  1. The authors should present criteria of determining the low, medium and high reputation incongruence.
  • The notes for the criteria were added in Figure 1 and 2 (Line 384 and 404)
  1. References to more current publications would be welcomed.
  • We added/replaced references to increase the recency. Following is the list of the new citations.
  • Chae, H., Song, J., & Lange, D. 2020. Basking in reflected glory: Reverse status transfer from foreign to home markets. Strategic Management Journal, n/a.
  • Din, S., Ishfaq, M., Khan, M. I., & Khan, M. A. 2019. A study of role stressors and job satisfaction: The case of mncs in collectivist context. Behavioral Sciences, 9(5): 49.
  • 2020. World’s Most Admired Companies, URL: https://fortune.com/worlds-most-admired-companies/
  • Gyan, A. K., Brahmana, R., & Bakri, A. K. 2017. Diversification strategy, efficiency, and firm performance: Insight from emerging market. Research in International Business and Finance, 42: 1103-1114.
  • Maclean, M., Harvey, C., & Clegg, S. R. 2016. Conceptualizing historical organization studies. Academy of Management Review, 41(4): 609-632.
  • Hill, C. W. L., & Rothaermel, F. T. 2003. The performance of incumbent firms in the face of radical technological innovation. Academy of Management Review, 28(2): 257-274.
  • Miller, S. R., Eden, L., & Li, D. 2020. CSR Reputation and Firm Performance: A Dynamic Approach. Journal of Business Ethics, 163(3): 619-636.
  • Polkovnichenko, V. 2005. Household portfolio diversification: a case for rank-dependent preferences. Review of Financial Studies, 18(4): 1467-1502.
  • Ravasi, D., Rindova, V., Etter, M., & Cornelissen, J. 2018. The formation of organizational reputation. Academy of Management Annals, 12(2): 574-599.
  • Rosen, S. 2002. Markets and diversity. American Economic Review, 92(1): 1-15.
  • Shinzato, T. 2017. Minimal investment risk of a portfolio optimization problem with budget and investment concentration constraints. Journal of Statistical Mechanics: Theory and Experiment, 2017(2): 023301.
  • Yue, P., Gizem Korkmaz, A., & Zhou, H. 2020. Household Financial Decision Making Amidst the COVID-19 Pandemic. Emerging Markets Finance and Trade, 56(10): 2363-2377.

  1. In line 242 the formula is not displayed correctly - the sum symbol is missing (this may be the result of converting the document to PDF)
  • We also guess this is due to a false file conversion and will work with editorial office.
  1. In line 258 the order of words is wrong - "employers and as students well as stakeholders", should be "employers and students as well as stakeholders" probably.
  • Thank you so much! Corrected (Line 265)
  1. After table 2, the line numbering is missing until the end of the text
  • We fixed the issue.

We hope our such effort reflected in the new draft could meet your expectation and look forward to discussing further.

Sincerely,

The corresponding author

Reviewer 2 Report

The article deals with the reputation of incongruence in case of MBA rankings. The article is methodically at a high level. However, it needs significant improvements

The goal is only analytical, and it is not clear from the title whether paper focused on the reputation of organizations or schools.

It would also help if the authors indicated for which organizations the replication of training of future employees is important (financial companies, industrial, or other?). In which countries is this important (in USA?)? Aren't the skills, abilities of employees that predict their success in employment more important?

How significant is the effect of MBA study reputation in terms of the company's overall reputation? Is it not more important that company politics or corporate culture is focused on corporate responsibility?

The most of the sources are more than 10 and more years old. Thus, authors should add current sources dealing with this issue. This is major problem of the paper.

How important will be reputation based on the MBA ranking in current pandemic situation (education is realised via online)?

I recommend add the final picture (figure) of model (with relationships) for better understanding of results.

Please, also add to the Conclusions limitations of your paper, your future research, and contribution of your approach to practice. It is still not clear from the current paper.

Author Response

Dear Reviewer 2,

We would like to thank you for the thorough and constructive review for the paper. It was also greatly appreciated to have a chance to improve the initial draft and resubmit the current version of the manuscript. Without your positive feedback and helpful comments, this would not be possible at all. The attached file is the revised manuscript in a Word format.

Let us address all the issues you pointed out with the resubmitted manuscript as follow:

  1. The goal is only analytical, and it is not clear from the title whether paper focused on the reputation of organizations or schools.
  • Thanks to your comment, we tried to clarify that the contribution, implication, and focus of our study is to generalize the findings to broader context. This effort was reflected in the resubmitted paper in introduction (Line 58 and 59) and conclusion (Line 432 to 496; Line 507 to 534). As we especially agreed with your point that the conclusion was the weakest compared to the methodology, we added five new paragraphs in the revised conclusion section.
  1. It would also help if the authors indicated for which organizations the replication of training of future employees is important (financial companies, industrial, or other?). In which countries is this important (in USA?)? Aren't the skills, abilities of employees that predict their success in employment more important?
  • Among the added paragraphs, Line 507 to 516 may directly address this point as the “automobile and pharmaceutical industries” as the empirical venues for the replication. When it comes to a geographic generalizability, we just wanted to give a sense that our findings were valid already since we used global data not restricting cases’ nationality.
  1. How significant is the effect of MBA study reputation in terms of the company's overall reputation? Is it not more important that company politics or corporate culture is focused on corporate responsibility?
  • To address your critical comment about the generalizability of our findings, we also decided to invite more theoretical arguments by comparing two conceptual framework: (1) portfolio theory versus (2) diversification theory (Line 432 to 496). With this new argument, we indirectly answered and reflected this point. Our focus on reputation is much closer to a strategic view to communicate with outside stakeholders, compared to internal views like organizational culture and politics. The future direction can cover this, so we suggested some internal view in the limitation part (Line 519 to 521), which is direct response to your concern.
  1. The most of the sources are more than 10 and more years old. Thus, authors should add current sources dealing with this issue. This is major problem of the paper.
  • Thanks a lot for the valid suggestion! We added/replaced references to increase the recency. Following is the list of the new citations.
  • Chae, H., Song, J., & Lange, D. 2020. Basking in reflected glory: Reverse status transfer from foreign to home markets. Strategic Management Journal, n/a.
  • Din, S., Ishfaq, M., Khan, M. I., & Khan, M. A. 2019. A study of role stressors and job satisfaction: The case of mncs in collectivist context. Behavioral Sciences, 9(5): 49.
  • 2020. World’s Most Admired Companies, URL: https://fortune.com/worlds-most-admired-companies/
  • Gyan, A. K., Brahmana, R., & Bakri, A. K. 2017. Diversification strategy, efficiency, and firm performance: Insight from emerging market. Research in International Business and Finance, 42: 1103-1114.
  • Maclean, M., Harvey, C., & Clegg, S. R. 2016. Conceptualizing historical organization studies. Academy of Management Review, 41(4): 609-632.
  • Hill, C. W. L., & Rothaermel, F. T. 2003. The performance of incumbent firms in the face of radical technological innovation. Academy of Management Review, 28(2): 257-274.
  • Miller, S. R., Eden, L., & Li, D. 2020. CSR Reputation and Firm Performance: A Dynamic Approach. Journal of Business Ethics, 163(3): 619-636.
  • Polkovnichenko, V. 2005. Household portfolio diversification: a case for rank-dependent preferences. Review of Financial Studies, 18(4): 1467-1502.
  • Ravasi, D., Rindova, V., Etter, M., & Cornelissen, J. 2018. The formation of organizational reputation. Academy of Management Annals, 12(2): 574-599.
  • Rosen, S. 2002. Markets and diversity. American Economic Review, 92(1): 1-15.
  • Shinzato, T. 2017. Minimal investment risk of a portfolio optimization problem with budget and investment concentration constraints. Journal of Statistical Mechanics: Theory and Experiment, 2017(2): 023301.
  • Yue, P., Gizem Korkmaz, A., & Zhou, H. 2020. Household Financial Decision Making Amidst the COVID-19 Pandemic. Emerging Markets Finance and Trade, 56(10): 2363-2377.
  1. How important will be reputation based on the MBA ranking in current pandemic situation (education is realised via online)?
  • We addressed this issue as a future direction (Line 514 to 516) while assuming the study’s finding is more generalizable than just about MBA ranking.
  1. I recommend add the final picture (figure) of model (with relationships) for better understanding of results.
  • We were able to add the Figure 3., thanks to your suggestion (Line 418 and 419) in the conclusion section.
  1. Please, also add to the Conclusions limitations of your paper, your future research, and contribution of your approach to practice. It is still not clear from the current paper.
  • We added conclusions with limitations, future direction, and practical values (Line 507 to 535).

We hope our such effort reflected in the new draft could meet your expectation and look forward to discussing further.

Sincerely,

The corresponding author

Reviewer 3 Report

Thank you very much for the opportunity to review this above titled paper. There are many things I like about the paper. First, the paper touches on an important subject in the literature which has been overlooked by researchers.

Although, the paper is very suitable to the requirement of Behavioral Sciences in many aspects, there are some very minor issues-

Abstract: These propositions about reputation incongruence is empirically examined in the context of MBA programs (are empirically examined)

Line 65-66: Fombrun et al. (1990; 2000) also conceptualize corporate reputation as a collective construct consisted of various attributes or dimensions. (also conceptualized; consisting of)

Line 109: As Ellsberg (1961) defines the ambiguity (defines ambiguity)

You can also see following article on responsible education-

Igwe, P. A., Rahman, M., Ohalehi, P., Amaugo, A., & Anigbo, J. A. (2020). Responsible education: what engages international postgraduate students–evidence from UK. Journal of Global Responsibility. Vol. 11 No. 4, pp. 363-376.

Author Response

Dear Reviewer 3,

We would like to thank you for the thorough and constructive review for the paper. It was also greatly appreciated to have a chance to improve the initial draft and resubmit the current version of the manuscript. Without your positive feedback and helpful comments, this would not be possible at all. The attached file is the revised manuscript in a Word format.

Let us address all the issues you pointed out with the resubmitted manuscript as follow:

  1. Abstract: These propositions about reputation incongruence is empirically examined in the context of MBA programs (are empirically examined)
  • We corrected the sentence (Line 17 and 18) thanks to your considerate review.
  1. Line 65-66: Fombrun et al. (1990; 2000) also conceptualize corporate reputation as a collective construct consisted of various attributes or dimensions. (also conceptualized; consisting of)
  • Corrected (Line 66).
  1. Line 109: As Ellsberg (1961) defines the ambiguity (defines ambiguity)
  • Corrected (Line 112).
  1. You can also see following article on responsible education- Igwe, P. A., Rahman, M., Ohalehi, P., Amaugo, A., & Anigbo, J. A. (2020). Responsible education: what engages international postgraduate students–evidence from UK. Journal of Global Responsibility. Vol. 11 No. 4, pp. 363-376.
  • We really appreciated the introduction of the reference. The article was included in the body of the revised manuscript, “The recent study of Igwe, Rahman, Ohalehi, Amaugo, and Anigbo (2020) also demonstrated that students in business schools perceived different value dimensions such as career development and academic learning to develop overall attitude to the programs.” (Line 83 to 85).

We hope our such effort reflected in the new draft could meet your expectation and look forward to discussing further.

Sincerely,

The corresponding author

Round 2

Reviewer 2 Report

Authors improved the paper text and it has significant impact on quality of the paper.

  1. The goal ..... Now are better explained efforts in paper.   OK
  2. The importance of training is now emphasized.   OK
  3. Significance of the MBA study effect. Theoretical contributions in conclusion part are sufficient.  OK
  4. References. I see overall improvement of resources mainly in the conclusion part. There is still possibility to use some of these new resources in introduction or literatury background.
  5. Importance of reputation is explained in future direction. OK
  6. Very good job with the figure 3!
  7. Limitation are added.   OK